# Volatile DMNT directly protects plants against *Plutella xylostella* by disrupting the peritrophic matrix barrier in insect midgut

**Chen Chen[1][†], Hongyi Chen[1][†], Shijie Huang[1][†], Taoshan Jiang[1], Chuanhong Wang[1], Zhen Tao[1], Chen He[1], Qingfeng Tang[2], Peijin Li[1]\***

[1]The National Key Engineering Lab of Crop Stress Resistance Breeding, the School of Life Sciences, Anhui Agricultural University, Hefei, China; [2]Key Laboratory of Biology and Sustainable Management of Plant Diseases and Pests of Anhui Higher Education Institutes, the School of Plant Protection, Anhui Agricultural University, Hefei, China

**Abstract** Insect pests negatively affect crop quality and yield; identifying new methods to protect crops against insects therefore has important agricultural applications. Our analysis of transgenic *Arabidopsis thaliana* plants showed that overexpression of *pentacyclic triterpene synthase 1*, encoding the key biosynthetic enzyme for the natural plant product (3E)-4,8-dimethyl-1,3,7-nonatriene (DMNT), led to a significant resistance against a major insect pest, *Plutella xylostella*. DMNT treatment severely damaged the peritrophic matrix (PM), a physical barrier isolating food and pathogens from the midgut wall cells. DMNT repressed the expression of *PxMucin* in midgut cells, and knocking down *PxMucin* resulted in PM rupture and *P. xylostella* death. A 16S RNA survey revealed that DMNT significantly disrupted midgut microbiota populations and that midgut microbes were essential for DMNT-induced killing. Therefore, we propose that the midgut microbiota assists DMNT in killing *P. xylostella*. These findings may provide a novel approach for plant protection against *P. xylostella*.

**\*For correspondence:** Peijin.li@ahau.edu.cn

[†]These authors contributed equally to this work

**Competing interests:** The authors declare that no competing interests exist.

## Introduction

Insect pests are one of the major factors affecting plant development and seed production, and cause tremendous global economic losses. Mitigating the detrimental effects of insect pests constitutes a long-term goal for researchers, breeders, and farmers (*Johnson and Züst, 2018*; *Tay and Gordon, 2019*). The diamondback moth *Plutella xylostella* (Lepidoptera: *Plutellidae*) has been ranked as one of the most difficult pests to control due to its short life cycle, high migration rate, strong tolerance to environmental stresses, and rising resistance to available insecticides (*Shakeel et al., 2017*; *Vaschetto and Beccacece, 2019*). An estimated 4–5 billion dollars are devoted each year to control of this pest worldwide (*Zalucki et al., 2012*).

Plants have evolved various defense mechanisms to protect themselves from insect pests. As part of their defenses, plants can kill or repel insects indirectly; for example, attacked plants emit volatile pheromones that attract insect predators to come and kill the initial pests (*Pichersky et al., 2006*; *Unsicker et al., 2009*; *Gols, 2014*; *Liu et al., 2018*). This strategy has been successfully adopted for plant protection in the field (*Turlings and Ton, 2006*; *Clavijo McCormick et al., 2012*; *Li et al., 2018*). However, pest management by genetic manipulation of volatiles in the plant is not always successful in the field due to its complex context-dependency (*Bruce et al., 2015*). In another line of defense, plants communicate danger signals to other plants (*Karban et al., 2006*; *Karban et al.,*

*2014*; *Kalske et al., 2019*). For example, the volatile homoterpene compound (3E)-4,8-dimethyl-1,3,7-nonatriene (DMNT) mediates plant–plant communication in sweet potato (*Ipomoea batatas*) and triggers systemic and jasmonic acid (JA)-independent anti-herbivore defenses in neighboring plants. These plants are induced to synthesize the protease inhibitor and storage protein sporamin, which is toxic to chewing pests (*Meents et al., 2019*). Similar results were also obtained in tea plants (*Camellia sinensis*) (*Jing et al., 2020*). Gasmi and colleagues reported that exposure of *Spodoptera exigua* larva to herbivore-induced plant volatiles (HIPVs) such as indole increased its susceptibility to *nucleopolyhedrovirus* and *Bacillus thuringiensis*, and additionally they found that the microbiota population in larval midgut was significantly affected, but the mechanism remains unclear (*Gasmi et al., 2019*). Another major line of plant defense relies on the ability to kill pests directly via the secretion of natural products that are harmful to insects. For instance, various HIPVs including indole have been reported to show direct toxicity to beet armyworm (*Maurya et al., 2020*). Some maize (*Zea mays* L.) ecotypes produce a 33 kDa cysteine protease in response to caterpillar feeding that damages the insect intestine, leading to death (*Pechan et al., 2002*). Several members of the Rubiaceae and Violaceae families produce cyclotides or macrocyclic peptides of 28–37 amino acids that disrupt epithelial cells in the midgut of cotton bollworm (*Helicoverpa armigera*) (*Barbeta et al., 2008*). Another effective defense compound is the secondary metabolite benzoxazinoids, which are produced in crops, provide resistance against a broad spectrum of pests, and are used as a resistance marker in crops (*Robert et al., 2017*). Although utilization of toxin variants or combinations of different toxins currently offer some protection against insects, the frequent emergence of insect resistance to many insecticides diminishes the effectiveness of these methods in the ongoing long-term battle to control insect pests (*Stokstad, 2001*; *Tay et al., 2015*; *Herrero et al., 2016*; *Naik et al., 2018*). Currently, reducing pest resistance has become a major issue in plant protection, and the identification of novel approaches is critical.

Mounting evidence shows that the insect midgut and the peritrophic matrix (PM) structure are the main targets of insecticides (*Kuraishi et al., 2011*; *Song et al., 2018*). The PM consists of a mixture of chitin and glycoproteins secreted by intestinal cells that serves as a first and essential barrier to isolate food and intestinal microbes from the intestinal cells of invertebrates such as *Bombyx mori*, *H. armigera*, *Anopheles gambiae*, and *Drosophila melanogaster* (*Hegedus et al., 2019*; *Liu et al., 2019*). The PM therefore prevents tissue infections in the midgut of insects (*Dinglasan et al., 2009*; *Erlandson et al., 2019*; *Hegedus et al., 2019*). In fruit fly (*D. melanogaster*), the protein drosocrystallin is a central component of the PM whose loss of function results in a reduction of PM width and an increase in its permeability, leading to higher sensitivity to bacterial infection from intestinal contents (*Kuraishi et al., 2011*). The mucin-like protein Mucin was reported to locate to the PM and plays important roles as a saliva component involved in the virulence, host adaptation, and immunity response of the brown planthopper (*Nilaparvata lugens*) (*Huang et al., 2017*; *Shangguan et al., 2018*). In maize, some cultivars develop a sharp epidermis on their leaf surface that can cause the PM of the fall armyworm (*Spodoptera frugiperda*) to rupture; this was proposed to be one of the mechanisms leading to pest death (*Mason et al., 2019*). In mosquito *Anopheles coluzzii*, the formation or maintenance of the PM depends on the midgut microbiota, and the PM ruptured when antibiotics were added to the blood meal and killed the microbiota (*Rodgers et al., 2017*). In agreement, several studies showed that mosquitoes became more susceptible to infections from the *Plasmodium* parasite or dengue viruses when their midgut microbiota was cleared, even though the exact causal mechanism remains elusive (*Dong et al., 2009*; *Wei et al., 2017*). Whether the intestinal microbiota has a direct role on PM formation in Lepidoptera has not been addressed.

While characterizing the function of pentacyclic triterpene synthase 1 (PEN1), an important enzyme catalyzing the production of the volatile metabolite DMNT in *Arabidopsis thaliana*, we demonstrate that PEN1 and its derived product DMNT play significant roles in pest control. Our data show that DMNT repels *P. xylostella*, eventually killing larvae by disrupting their PM. We establish that the midgut microbiota is required for this process, and that DMNT damages the PM by repressing the expression of *PxMucin*.

## Results

### DMNT repels and kills *P. xylostella* larvae

While investigating the function of the *A. thaliana* gene *PEN1* (at4g15340), we noticed that *PEN1*-overexpressing transgenic plants exhibited strong resistance to *P. xylostella* infestation (*Figure 1A, B*), and the *P. xylostella* larvae fed on *35Spro:PEN1* transgenic plants showed significantly lower survival rate than those on the non-transgenic control (*Figure 1C*). PEN1 has been reported to be a key enzyme in the biosynthesis of the volatile homoterpene DMNT (*Sohrabi et al., 2015*), which was confirmed by our gas chromatography-mass spectrometry (GC-MS) analysis on *35Spro:PEN1* transgenic plants. DMNT was enriched and could be highly induced by *P. xylostella* infestation (*Figure 1—figure supplement 1A–D*), suggesting that DMNT might play a role in plant resistance against this chewing insect. To validate this hypothesis, we synthesized DMNT and sprayed the chemical on *Brassica napus* leaves before feeding of *P. xylostella* larvae; this bioassay demonstrated that much less of the DMNT-treated leaves was eaten compared to the control leaves (*Figure 1D*). To determine if *P. xylostella* larvae sensed DMNT and avoided the volatile chemical, we set up an insect preference test (*Figure 1—figure supplement 2*). We placed forage containing DMNT or paraffin oil as control at either end of a glass test tube and positioned fourth-instar *P. xylostella* larvae in the center of the tube. As shown in *Figure 1E*, significantly more larvae moved away from the DMNT side after 10–120 min, indicating that *P. xylostella* larvae perceived and avoided the DMNT (*Figure 1E*).

We next investigated whether this compound was effective at killing pests directly by feeding second-instar larvae with forage containing DMNT or paraffin oil as control and scoring survival rates. Indeed, exposure to DMNT for 48 hr resulted in the death of ~20% of larvae, which increased to 50% after 120 hr; meanwhile, the larva growth and pupation rates were also severely affected (*Figure 1F–I*). To assess the potential dose dependence of DMNT-mediated lethality on *P. xylostella* larvae, we allowed larvae to feed on forage containing 0.7, 7, or 70 μM DMNT. Although the overall survival and pupation rates of *P. xylostella* larvae significantly decreased at all concentrations, we observed no significant effect of dosage (*Figure 1—figure supplement 3A, B*), in agreement with results obtained with the root-rot pathogen *Pythium irregulare* (*Sohrabi et al., 2015*). Considering the fact that DMNT in *A. thaliana* plants can be continuously induced by *P. xylostella* infestation, accumulating to ~70 ng/g after 3–21 hr (*Figure 1—figure supplement 1C, D*), and in contrast DMNT injected in forage for bioassays will become less due to volatilizing along with the process of experiments, we selected the dosage of 7 μM DMNT for further studies.

We then tested the sensitivity of larvae of different ages to DMNT treatment. Accordingly, we fed first- to fourth-instar larvae with forage containing DMNT, which revealed that DMNT significantly affected all stages of *P. xylostella* larvae tested in these assays, although first- and second-instar larvae had a more pronounced response relative to other stages (*Figure 1—figure supplement 4*). First-instar *P. xylostella* larvae were very delicate and showed very low pupation and eclosion rates of ~65% even in the controls, making them unsuitable for mechanistic studies (*Figure 1—figure supplement 4A–D*, left-side row). We therefore chose second-instar larvae as a test system for subsequent experiments.

*P. xylostella* larvae appeared to display significantly lower forage intake and less defecation when treated with DMNT (*Figure 1—figure supplement 5A–D*), raising the possibility that *P. xylostella* larvae might be dying from starvation. To test this hypothesis, we compared the mortality rates of larvae challenged with either DMNT treatment or starvation (by withholding forage feeding) as a time course. As shown in *Figure 2A*, DMNT-treated *P. xylostella* larvae died at least 12 hr earlier than those experiencing starvation. In addition, starved larvae had much smaller bodies at the time of death relative to DMNT-treated larvae (*Figure 1—figure supplement 5E*), indicating that *P. xylostella* larvae were likely quickly poisoned to death by DMNT rather than being killed by DMNT-induced starvation. In light of the growth inhibition of *P. xylostella* larvae by DMNT (*Figure 1*, *Figure 1—figure supplement 4*), we hypothesized that the larvae had a reduced ability to digest food in gut and isolated larval midguts after a 48 hr exposure to DMNT to assay the activity of the digestive enzyme lipase, an indicator for digestive capacity (*Santana et al., 2017*; *Hu et al., 2019*). When compared to the controls, DMNT-treated larvae showed significantly lower lipase activity, suggesting that their gut function was indeed disrupted (*Figure 2B*).

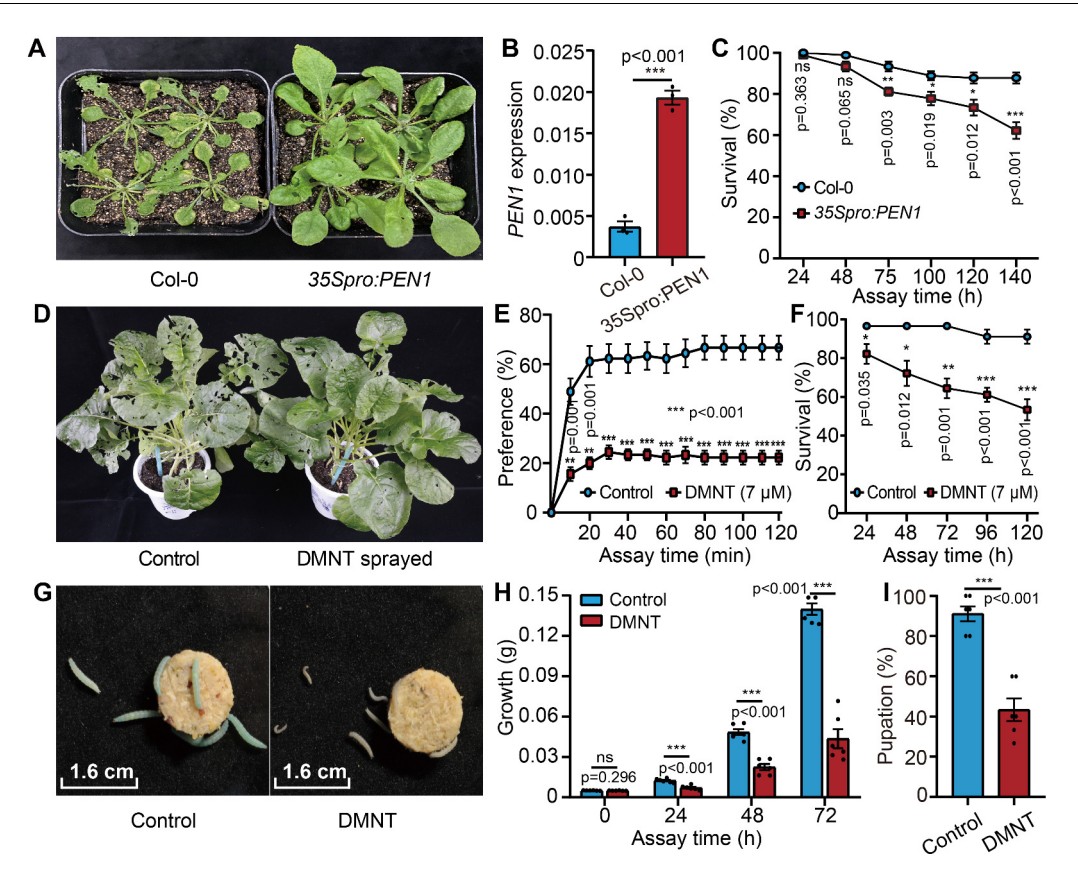

**Figure 1.** (3E)-4,8-Dimethyl-1,3,7-nonatriene (DMNT) repels and kills *P. xylostella* larvae. (**A**) Pentacyclic triterpene synthase 1 (*PEN1*) overexpression in *A. thaliana* results in high resistance to *P. xylostella* infestation. *PEN1* encodes the key enzyme responsible for DMNT biosynthesis in *Arabidopsis* plants. (**B**) *PEN1* is overexpressed in *35Spro: PEN1* transgenic *Arabidopsis* plants. (**C**) *35Spro:PEN1* transgenic *Arabidopsis* plants cause lower survival of *P. xylostella* than wild type control. (**D**) *B. napus* plants sprayed with DMNT show strong resistance to *P. xylostella* larvae. (**E**) *P. xylostella* larvae can sense and are repelled by DMNT. The preference test system is illustrated in *Figure 1—figure supplement 2*. (**F**) DMNT treatment significantly lowers the survival of *P. xylostella* larvae. (**G**) Phenotypic comparison between DMNT-fed larvae and control larvae (treated with the DMNT solvent paraffin oil). (**H**) Growth inhibition of *P. xylostella* larvae by DMNT. (**I**) Reduced pupation of *P. xylostella* larvae by DMNT treatment. Error bars represent the standard error of six independent biological replicates, with 15 larvae per replicate. Asterisks indicate significant differences (*p<0.05, **p<0.01, ***p<0.001, ns, not significant; two-tailed unpaired *t*-test).

The online version of this article includes the following source data and figure supplement(s) for figure 1:

**Source data 1.** (3E)-4,8-Dimethyl-1,3,7-nonatriene (DMNT) repels and kills *P. xylostella* larvae.

**Figure supplement 1.** (3E)-4,8-Dimethyl-1,3,7-nonatriene (DMNT) is enriched in *35Spro:PEN1* transgenic *A. thaliana* plants and can be highly induced by *P. xylostella* infestation.

**Figure supplement 1—source data 1.** (3E)-4,8-Dimethyl-1,3,7-nonatriene (DMNT) is enriched in *35Spro:PEN1* transgenic *A. thaliana* plants and can be highly induced by *P. xylostella* infestation.

**Figure supplement 2.** A schematic representation of the principle behind the insect choice test system.

**Figure supplement 3.** (3E)-4,8-Dimethyl-1,3,7-nonatriene (DMNT) shows a slight dosage-dependent influence on the survival and pupation rates of *P. xylostella* larvae.

**Figure supplement 3—source data 1.** (3E)-4,8-Dimethyl-1,3,7-nonatriene (DMNT) shows a slight dosage-dependent influence on the survival and pupation rates of *P. xylostella* larvae.

**Figure supplement 4.** (3E)-4,8-Dimethyl-1,3,7-nonatriene (DMNT) treatment affects *P. xylostella* larvae of different ages.

**Figure supplement 4—source data 1.** (3E)-4,8-Dimethyl-1,3,7-nonatriene (DMNT) treatment affects *P. xylostella* larvae of different ages.

*Figure 1 continued on next page*

**Figure supplement 5.** (3E)-4,8-Dimethyl-1,3,7-nonatriene (DMNT) treatment lowers forage intake and defecation of *P. xylostella* larvae.

**Figure supplement 5—source data 1.** (3E)-4,8-Dimethyl-1,3,7-nonatriene (DMNT) treatment lowers forage intake and defecation of *P. xylostella* larvae.

## DMNT treatment damages the midgut barrier of *P. xylostella* larvae

To determine the possible cause of pest death brought upon by DMNT, we carried out the 'Smurf test' to assay leakage problems in the midgut of insects. This test uses the blue dye erioglaucine disodium salt; the molecular weight of this dye is 792.85, too big to pass passively through the intestinal barrier, making it a useful tool to reveal the status of epithelium lining in the intestines (*Amcheslavsky et al., 2009*; *Rera et al., 2012*; *He et al., 2017*). We added the dye to forage,

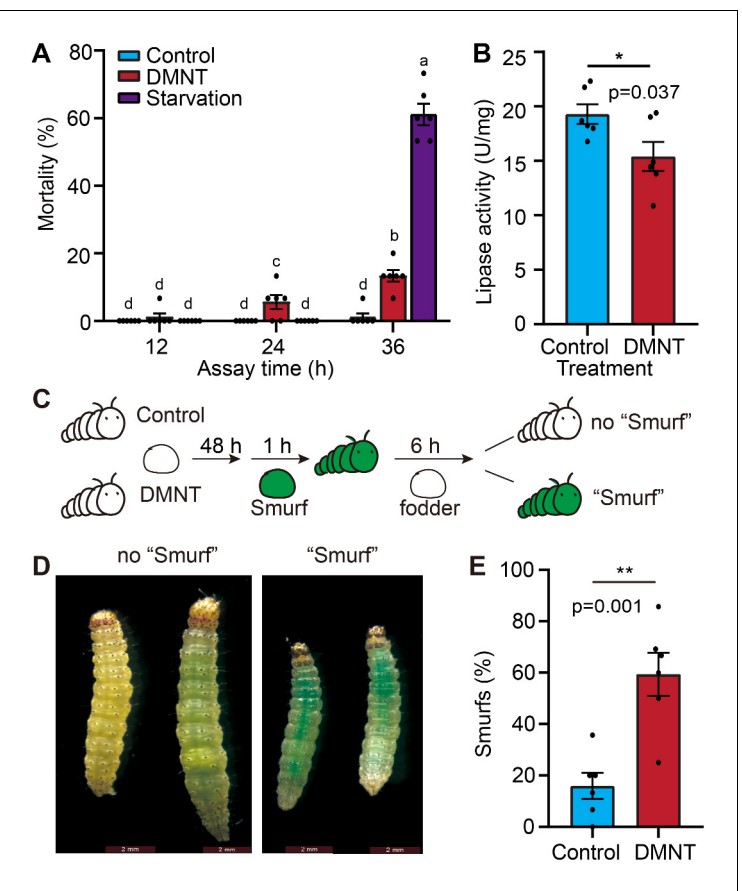

**Figure 2.** The midgut barrier of *P. xylostella* larvae is damaged by (3E)-4,8-dimethyl-1,3,7-nonatriene (DMNT) treatment. (**A**) Larvae die earlier from treatment with DMNT than from starvation. (**B**) Decreased lipase activity in the midgut of *P. xylostella* larvae upon DMNT treatment. (**C**) The principle of the Smurf test. Larvae are fed with forage loaded with the dye erioglaucine disodium salt for 1 hr before returning to normal forage without dye. After 6 hr, the extent of dye retention is monitored. (**D**) Representative images of DMNT-treated larvae showing dye retention, as evidenced by their blue appearance, like the Smurf cartoon character. (**E**) Quantification of results shown in (**D**). The different letters in (**A**) indicate a significant difference (one-way ANOVA). Error bars in (**B**, **E**) represent the standard error of six independent biological replicates, with 15 larvae per replicate. Asterisks in (**B**, **E**) indicate significant differences (*$p<0.05$, **$p<0.01$, ns, not significant; two-tailed unpaired *t*-test). The online version of this article includes the following source data for figure 2:

**Source data 1.** The midgut barrier of *P. xylostella* larvae is damaged by (3E)-4,8-dimethyl-1,3,7-nonatriene (DMNT) treatment.

allowed larvae to feed for 1 hr, before returning them to forage without dye for 6 hr (*Figure 2C*). At the end of the feeding period, control larvae had defecated the dye and showed no obvious accumulation of the dye in their body. In contrast, DMNT-treated larvae retained significant amounts of dye in the midgut and the surrounding tissues (*Figure 2D*), effectively turning the larvae blue, like the Smurf cartoon character. A quantitative assessment of blue (Smurf) and normal (no Smurf) larvae showed that DMNT treatment resulted in three times as many Smurf larvae as no Smurf larvae (*Figure 2E*). These results suggested that DMNT caused lesions at the larval midgut and resulted in higher permeability.

## The PM is damaged by DMNT

We dissected and isolated larval PM from control and DMNT-treated larvae to perform a comparative analysis of the structure under a stereomicroscope. After DMNT treatment for 48 hr, the PM became thin and loose, whereas the PM of control samples remained intact and plump (*Figure 3A, B*). To further document the observed midgut damages, we performed transverse sections on larvae treated with DMNT for 24–48 hr. Hematoxylin-eosin (HE) staining revealed that the PM of control larvae was thick and intact (*Figure 3C, E*), whereas the PM of larvae treated with DMNT for 24 hr was thin and discontinuous in some regions (*Figure 3D*). This effect became more pronounced after 48 hr of exposure to DMNT as the PM was completely corrupted and the intestinal

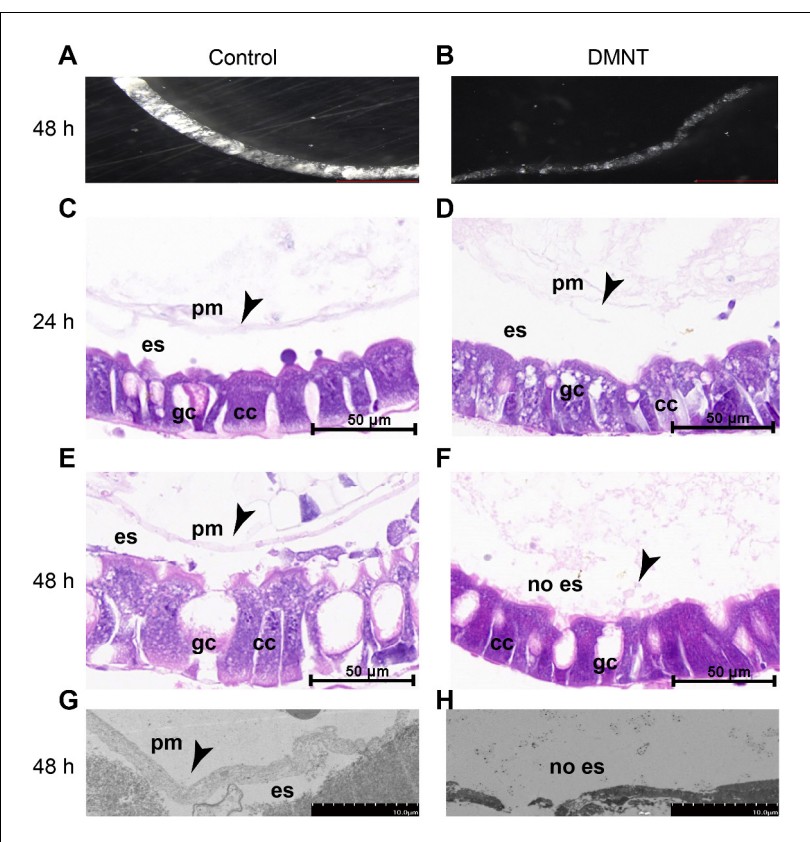

**Figure 3.** The peritrophic matrix (PM) structure is damaged by (3E)-4,8-dimethyl-1,3,7-nonatriene (DMNT) treatment. (**A, B**) PM ultrastructure of control (**A**) and DMNT-fed (**B**) *P. xylostella* larvae for 48 hr. Note the thin and delicate PM in DMNT-fed larvae. (**C–F**) Transverse section and hematoxylin-eosin staining show damage of the PM by DMNT after exposure for 24 hr (**C, D**) and 48 hr (**E, F**). (**G, H**) The PM is damaged by DMNT feeding, as shown by transmission electron microscopy. Arrowheads indicate the PM. es: delimits the ectoperitrophic space; gc: goblet cells; cc: columnar cells.

The online version of this article includes the following source data for figure 3:

**Source data 1.** The peritrophic matrix (PM) structure is damaged by (3E)-4,8-dimethyl-1,3,7-nonatriene (DMNT) treatment.

content got close to the intestine wall, with several enterocyte cells spilling from the epithelial wall (*Figure 3F*). We validated these results by transmission electron microscopy (*Figure 3G, H*).

## DMNT represses the expression of the mucin-like gene *PxMucin*

To investigate the molecular mechanism underlying the function of DMNT function on *P. xylostella* larvae, we performed a whole transcriptome sequencing (RNA-seq) analysis. Compared with the control, after 12 hr DMNT treatment, there were 189 genes differentially expressed with a minimum cutoff of twofold change; among these 189 genes, 62 were upregulated and 127 were downregulated (*Supplementary file 1*). Enrichment analysis at the Kyoto Encyclopedia of Genes and Genomes indicated that these differentially expressed genes belonged to more than 20 different metabolic pathways, including tyrosine, purine and carbon metabolism, proteasome, wax, and longevity regulation, which coincide with the developmental defects of *P. xylostella* larvae (*Figure 4—figure supplement 1*). A gene *PxMucin*, encoding a mucin-like protein, caught our attention as its protein was reported to accumulate in the PM (*Sarauer et al., 2003*; *Israni and Rajam, 2017*; *Figure 4A*). After DMNT treatment for 48 hr, *PxMucin* transcript levels were significantly downregulated compared to control larvae (*Figure 4B, C*). We then synthesized double-stranded RNA (dsRNA) designed against *PxMucin* using an in vitro transcription technique and fed *P. xylostella* larvae with *A. thaliana* leaves coated with this dsRNA to induce RNA interference (RNAi) of the target gene in the insect gut (*Figure 4A*, *Supplementary file 2*). This dsRNA significantly downregulated endogenous *PxMucin* in the dsRNA-fed larvae (*Figure 4D, E*) and resulted in a significant reduction in growth, survival, and pupation rates, as well as lighter larval bodies (*Figure 4F–I*). Furthermore, transverse sections of dsRNA-fed larvae indicated that the PM structure became thin and disappeared in some regions (*Figure 4J, K*), observations that were similar to the DMNT-treated larva (*Figure 3*), suggesting that DMNT may disrupt the PM partially through the repression of *PxMucin* expression.

In addition, to check if the dsRNA treatment downregulates other possible genes except for *PxMucin*, we used the amino acid sequence of PxMucin to search for its homologs from NCBI protein database and detected three genes, *peritrophin 1 like* (LOC105381868), *chondroitin proteoglycan 2 like* (LOC119693213), and *mucin 2 like* (LOC105395714), which have 37%, 55%, and 36% protein homology with PxMucin, respectively. The followed qRT-PCR analysis showed that there was no significant difference in the expression levels of these three genes between the dsRNA-treated and non-treated control (*Figure 4—figure supplement 2*). These results suggest that the dsRNA against *PxMucin* likely has low off-target, and the developmental defects of the dsRNA-treated *P. xylostella* larvae were mainly caused by the downregulation of *PxMucin*.

## Gut microbes are essential for DMNT function

Previous research reported that the microbiota of the mosquito midgut is essential for the maintenance of PM integrity (*Rodgers et al., 2017*). To test whether antibiotics have similar effect on *P. xylostella* PM and larva development, we added an antibiotic cocktail into forage, which effectively decreased the midgut microbiota (*Figure 5—figure supplement 1*). We therefore fed second-instar larvae with forage laced with antibiotics. In contrast to the results obtained in mosquito, antibiotic treatment had no significant influence on the survival or pupation rates of *P. xylostella* larvae (*Figure 5A, B*). Consistent with this observation, transverse section and HE staining indicated that the PM was intact even after the removal of the midgut microbiota by the antibiotic cocktail, suggesting that the formation and/or maintenance of the *P. xylostella* PM are not dependent on the midgut microbiota (*Figure 5—figure supplement 2*).

To test whether midgut microbes contribute to DMNT-induced insect death, we next exposed larvae to a combination of antibiotics and DMNT. Our results indicated that the effects of DMNT on *P. xylostella* were completely abolished upon the addition of antibiotics (*Figure 5C, D*, *Figure 5—figure supplement 1*). These results also raised the possibility that the loss of DMNT toxicity following antibiotic treatment extended to protection of the PM structure against DMNT-induced damage. We therefore carried out transverse sections and HE staining of treated larvae to test this hypothesis. Quite on the contrary, the midgut section showed a similarly damaged PM structure upon treatment with either DMNT alone, or with the combination of DMNT and antibiotics, suggesting that an active population of gut microbiota contributes to the DMNT-induced PM damage that kills larvae, but that gut microbiota do not directly affect the PM (*Figure 5E–H*). In addition, we determined that

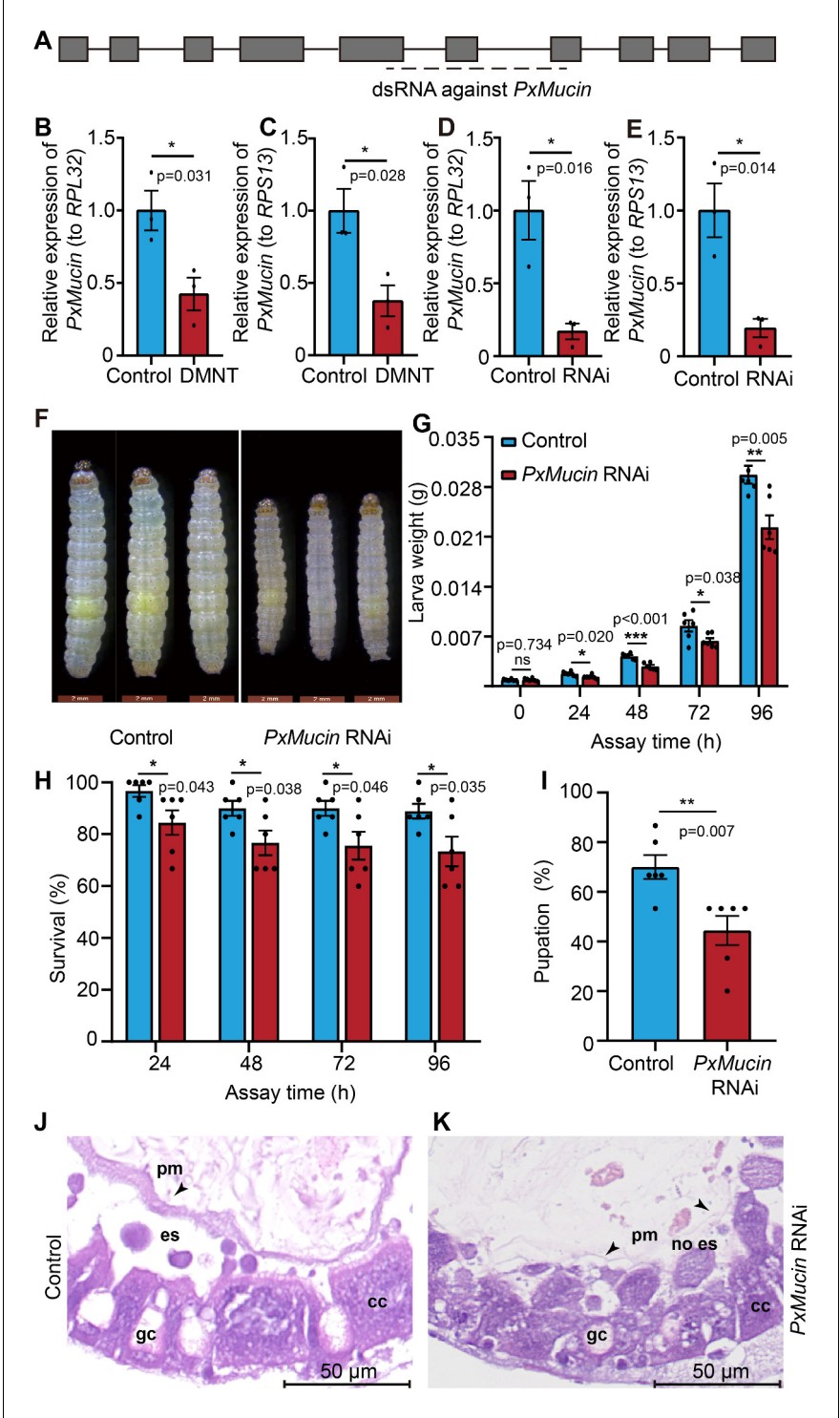

**Figure 4.** (3E)-4,8-Dimethyl-1,3,7-nonatriene (DMNT) disrupts the peritrophic matrix (PM) structure of *P. xylostella* larvae via repression of the mucin-like gene *PxMucin*. (**A**) Gene structure of *PxMucin*. Boxes represent exons, and horizontal lines represent introns. The horizontal dashed line indicates the targeted regions by double-stranded RNA (dsRNA). (**B, C**) DMNT treatment downregulates the expression of *PxMucin*. (**D, E**) Successful RNA interference (RNAi) of *PxMucin* by dsRNA feeding. (**F–I**) Larvae fed with dsRNA against *PxMucin* show impaired body development (**F**), weight (**G**), survival (**H**), and pupation (**I**) rates. (**J, K**) PM structure from control larvae (**J**) and larvae-fed *PxMucin* dsRNA (**K**). Note how *PxMucin* dsRNA feeding phenocopies DMNT treatment. Error bars in (**B–E**) represent the standard error of three independent biological replicates, with 30 *P. xylostella* larvae per

*Figure 4 continued on next page*

*Figure 4 continued*

replicate. Error bars in (**G–I**) represent the standard error of six independent biological replicates, with 15 *P. xylostella* larvae per replicate. Asterisks indicate significant differences (*p<0.05, **p<0.01, ***p<0.001, ns, not significant; two-tailed unpaired *t*-test).

The online version of this article includes the following source data and figure supplement(s) for figure 4:

**Source data 1.** (3E)-4,8-Dimethyl-1,3,7-nonatriene (DMNT) disrupts the peritrophic matrix (PM) structure of *P. xylostella* larvae via repression of the mucin-like gene *PxMucin*.

**Figure supplement 1.** Transcriptomic analysis of the differentially expressed genes in the *P. xylostella* larva after (3E)-4,8-dimethyl-1,3,7-nonatriene (DMNT) treatment at 12 hr.

**Figure supplement 2.** The expression of three *PxMucin* homologs is not significantly influenced by the double-stranded RNA (dsRNA) treatment.

**Figure supplement 2—source data 1.** The expression of three *PxMucin* homologs is not significantly influenced by the double-stranded RNA (dsRNA) treatment.

the midgut microbiota could pass the PM barrier and invade midgut cells more easily when the PM was damaged by DMNT compared to controls (*Figure 5I, J*).

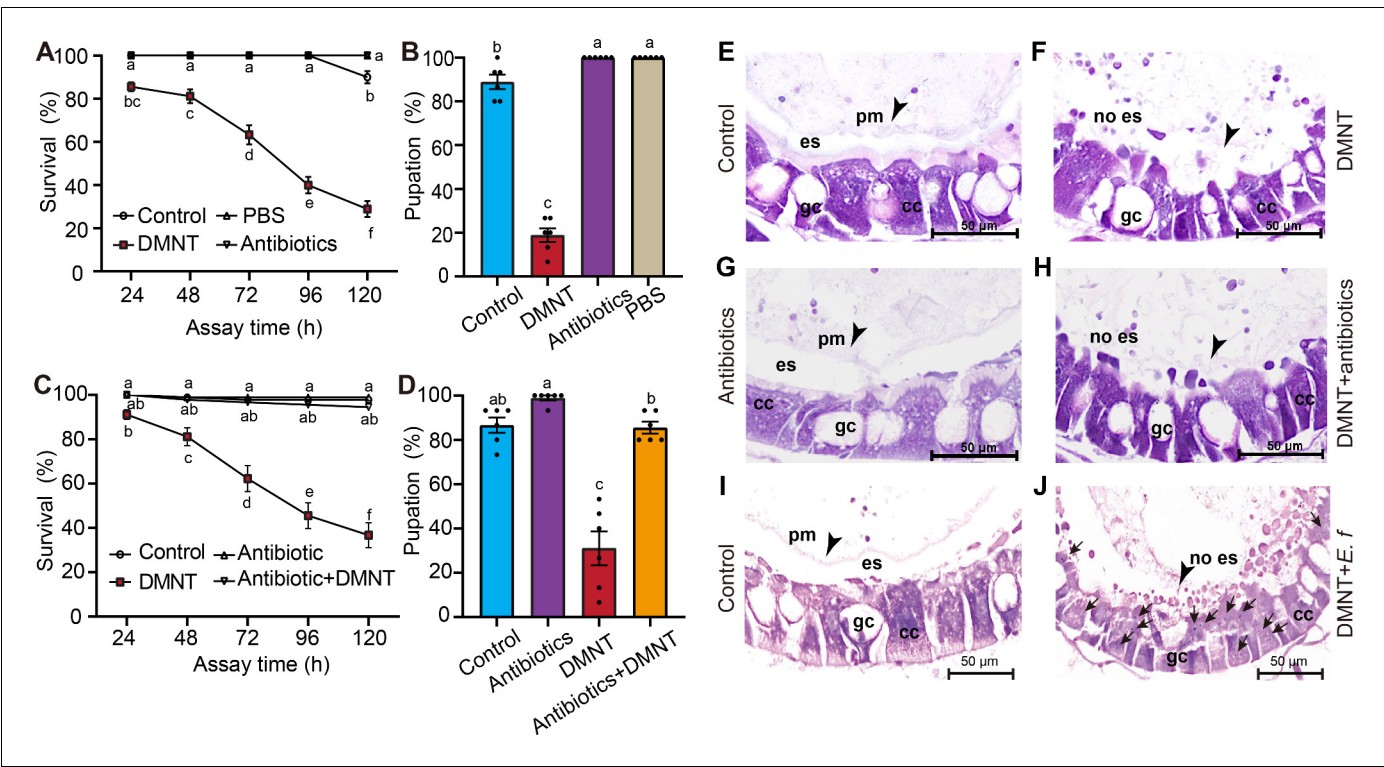

**Figure 5.** Microbes in the midgut are indispensable for (3E)-4,8-dimethyl-1,3,7-nonatriene (DMNT)-mediated killing of *P. xylostella* larvae. (**A, B**) Removal of the microbiota by antibiotics alone has no influence on larval survival (**A**) or pupation rates (**B**). DMNT treatment was used as a control. (**C, D**) Removal of the microbiota by treatment with antibiotics eliminates the adverse DMNT-mediated effects on the survival (**C**) and pupation (**D**) rates of larvae. (**E–H**) Comparison of peritrophic matrix (PM) structure in control larvae (**E**), larvae treated with DMNT alone, (**F**) or in combination with antibiotics (**H**). Note the disruption of the PM, whereas treatment with the antibiotic cocktail alone has no effects (**G**). (**I, J**) *Enterococcus faecalis* (*E. f*) can invade midgut cells when the PM is disrupted by DMNT treatment (**J**), but is restricted within the PM in controls (**I**). Arrows indicate *E. f* cells stained with dye. Arrowheads indicate the PM. es: delimits the ectoperitrophic space; gc: goblet cells; cc: columnar cells. The different letters in (**A–D**) indicate a significant difference (one-way ANOVA).

The online version of this article includes the following source data and figure supplement(s) for figure 5:

**Source data 1.** Microbes in the midgut are indispensable for (3E)-4,8-dimethyl-1,3,7-nonatriene (DMNT)-mediated killing of *P. xylostella* larvae.

**Figure supplement 1.** The midgut microbiota is completely killed by the antibiotic cocktail.

**Figure supplement 2.** Treatment with antibiotics does not influence the integrity of peritrophic matrix (PM) structures.

**Figure supplement 2—source data 1.** Treatment with antibiotics does not influence the integrity of peritrophic matrix (PM) structures.

## DMNT causes an imbalance in the gut microbe populations

We further investigated the composition of the internal microbiota within the midgut system upon DMNT treatment. A principal component analysis of midgut microbes based on 16S ribosomal DNA sequencing indicated that the DMNT treatment significantly disturbed the microbiota composition in the midgut of *P. xylostella* larvae (*Figure 6A, B*, *Figure 6—figure supplement 1*), with the abundance of 18 phyla, such as proteobacteria and firmicutes, showing strong changes after DMNT treatment (*Figure 6A*). As an example, the firmicutes bacterium *Enterococcus* was enriched around 85-fold in DMNT-treated larvae relative to the control (*Figure 6B*, *Figure 6—figure supplement 2*, *Supplementary file 3*).

To further explore the role of the midgut microbiota in the context of DMNT-induced killing of *P. xylostella* larvae, we performed additional experiments such as providing gut microbes in forage as a pathogen. To prepare second-instar 'antibiotics-treated' *P. xylostella* larvae, we inoculated larval eggs on plastic paper and put it close to forage containing antibiotics to ensure the larvae could eat antibiotics immediately after hatching; we found that when the larvae grew to second instar stage, the microbiota in gut were nearly killed by the antibiotic cocktail (*Figure 6—figure supplement 3*). Without DMNT, gut microbe feeding significantly decreased the survival or pupation rates of the larvae compared with the antibiotics-treated controls, and strikingly the addition of gut microbes alongside DMNT strongly enhanced the effects of DMNT treatment, all larvae died at 120 hr, and no larvae pupated in the end (*Figure 6C, D*). In addition, we detected the gene expression level of an immune factor lysozyme. RT-PCR results showed that the *lysozyme* expression of larvae gut was significantly downregulated from 48 to 72 hr after DMNT treatment, suggesting that DMNT-treated larvae had a weakened ability to resist pathogen invasions (*Figure 6—figure supplement 4*).

## Discussion

The insect *P. xylostella* is a devastating agricultural pest that causes tremendous economic loss globally and drives an excessive use of chemical insecticides; moreover, it is increasingly exhibiting resistance to these insecticides (*Tay and Gordon, 2019*). Thus, the identification of natural insecticides is very important for sustainable agriculture. In this study, we present strong evidence that the natural plant compound DMNT can serve as an effective bioinsecticide against *P. xylostella*. DMNT exposure damages the PM structure in a microbiota-assisted manner, partially by repressing the expression of *PxMucin*.

Increasing studies showed that plant volatiles can enhance the resistance of adjacent plants. For instance, tomatoes release green leaf alcohol (Z)-3-hexenol after insect infestation, which are converted into glycosides after being absorbed by neighboring plants, and cause direct damages to insects (*Sugimoto et al., 2014*). Another plant volatile indole was reported to show direct toxicity to insects, and it can regulate the balance between insects, their natural enemies, and host plants (*Veyrat et al., 2016*; *Ye et al., 2018*). DMNT exists in various plant species, and different aspects of this compound have been studied, such as DMNT biosynthesis in plants, as well as that DMNT plays a role in signal transmission to neighbor plants after being attacked by insects (*Sohrabi et al., 2015*; *Richter et al., 2016*; *Liu et al., 2018*), but these studies largely focused on its indirect role in plant protection (*Mumm et al., 2008*; *Li et al., 2018*; *Liu et al., 2018*; *Meents et al., 2019*; *Jing et al., 2020*). Indeed, whether (and how) DMNT plays a direct role in killing pests has remained unknown until now. Our data demonstrate that DMNT affects *P. xylostella* pests both in indirect and direct ways. The preference test indicated that DMNT is a volatile that can be sensed by *P. xylostella* larvae and repels these larvae (*Figure 1E*). This characteristic is critical and suggests that DMNT can be used as an insect repellent, driving pests away from plants following application.

In addition to repelling *P. xylostella*, DMNT treatment also significantly negatively affected *P. xylostella* growth (*Figure 1—figure supplement 5E*) and eventually led to lower survival and pupation rates (*Figure 1F*, *Figure 1I*, *Figure 1—figure supplement 4*), suggesting that this compound has both short- and long-term effects on pest development and reproduction. Meents and colleagues reported that the growth of *Spodoptera litura* showed no significant change after feeding with DMNT (20 µL, 1 mg/mL) for 10 days (*Meents et al., 2019*). In our study, as shown in *Figure 1—figure supplement 4*, the older instar larvae with larger body weight showed relatively higher tolerance to DMNT treatment. We hypothesize that probably *S. litura* is more tolerant to DMNT because it has quite a large body size, thus no clear effect was observed in Meents' study. In mosquitoes, PM

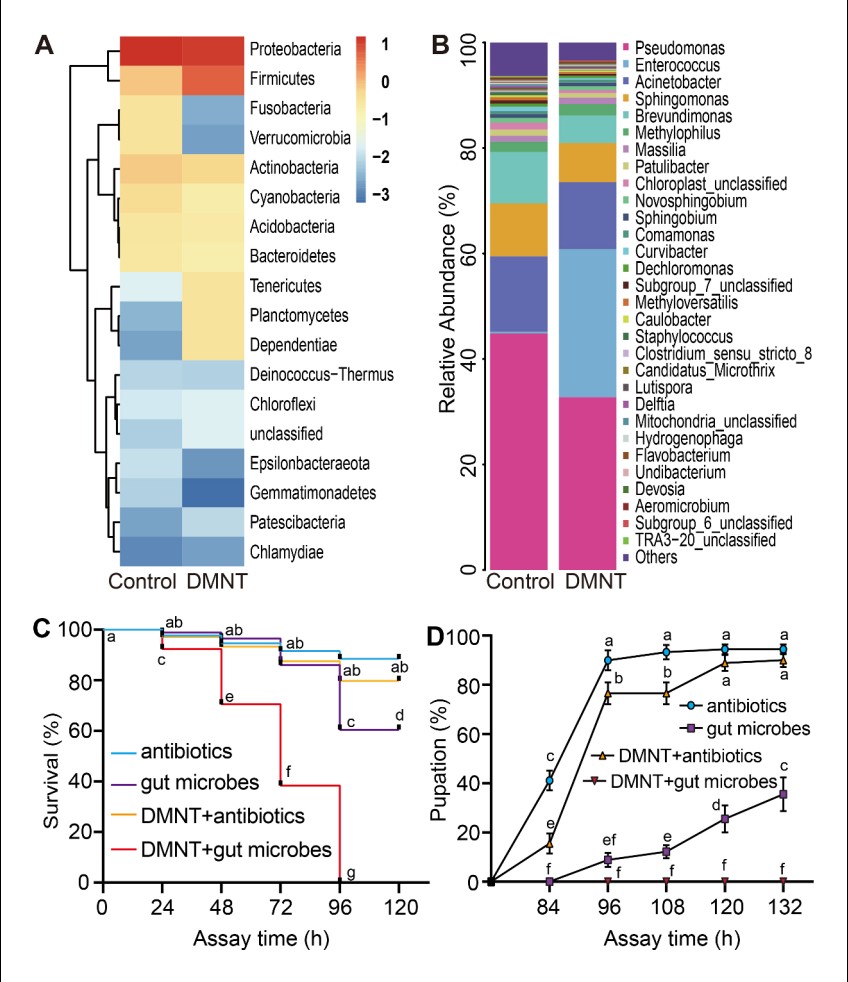

**Figure 6.** The microbiota composition of *P. xylostella* larvae midgut is affected by (3E)-4,8-dimethyl-1,3,7-nonatriene (DMNT) treatment. (**A**) Relative abundance of 18 phyla in response to DMNT treatment. (**B**) Effect of DMNT treatment on *Enterococcus* microbial populations. (**C, D**) Feeding with gut microbes decreases the survival (**C**) and pupation (**D**) rates of larvae treated with DMNT. Error bars represent the standard error of six independent biological replicates, with 15 *P. xylostella* larvae each. The different letters in (**C, D**) indicate a significant difference (one-way ANOVA).

The online version of this article includes the following source data and figure supplement(s) for figure 6:

**Source data 1.** The microbiota composition of *P. xylostella* larvae midgut is affected by (3E)-4,8-dimethyl-1,3,7-nonatriene (DMNT) treatment.

**Figure supplement 1.** (3E)-4,8-Dimethyl-1,3,7-nonatriene (DMNT) affects the abundance of microbiota populations in the midgut of *P. xylostella* larvae.

**Figure supplement 2.** *Enterococcus* sp. was enriched around 85-fold in (3E)-4,8-dimethyl-1,3,7-nonatriene (DMNT)-treated larvae than the control larvae.

**Figure supplement 2—source data 1.** *Enterococcus* sp. was enriched around 85-fold in (3E)-4,8-dimethyl-1,3,7-nonatriene (DMNT)-treated larvae than the control larvae.

**Figure supplement 3.** The gut microbiota from second-instar *P. xylostella* larvae were nearly killed by the antibiotic cocktail.

**Figure supplement 3—source data 1.** The gut microbiota from second-instar *P. xylostella* larvae were nearly killed by the antibiotic cocktail.

**Figure supplement 4.** *PxLysozyme* expression was downregulated from 48 to 72 hr.

**Figure supplement 4—source data 1.** *PxLysozyme* expression was downregulated from 48 to 72 hr.

integrity appears to require an intact midgut microbiota; this can be compromised by antibiotic treatment, leading to PM damage and higher susceptibility to infection (*Rodgers et al., 2017*). Our data show that the midgut microbiota is essential for DMNT function as its complete removal with antibiotics eliminates DMNT-induced larval death. However, in contrast to mosquitoes, the microbiota itself does not influence the PM structure or *P. xylostella* development (*Figure 5A–D, E, G*; *Rodgers et al., 2017*), suggesting that the midgut microbiota may play an assistant, but nevertheless essential, role in helping DMNT kill *P. xylostella*, similar to studies of the insecticide peptides from *B. thuringiensis* (*Caccia et al., 2016*).

We propose the following model: upon damage incurred by the PM in response to DMNT exposure (*Figure 3*), pathogens residing in the midgut pass through the PM lesions and invade midgut epithelial cells, leading to infection and tissue inflammation. In support of this hypothesis, we detected more pathogens in the midgut cells of larvae treated with DMNT in transverse sections (*Figure 5I, J*), and the lipase activity in DMNT-treated larvae was significantly reduced (*Figure 2B*). Furthermore, the analysis of microbiota populations showed that DMNT treatment severely affected the midgut environment, causing a reduction in the dominant bacterial populations seen in controls, whereas *Enterococcus* populations substantially increased (*Figure 6A, B*, *Figure 6—figure supplement 2*). Several terpenoid compounds have recently been reported to selectively modulate the *Arabidopsis* root microbiota (*Huang et al., 2019*). It suggests that plants may regulate microbial communities by releasing volatile compounds. Consistently, in the present study, we also found that DMNT has a significant influence on gut microbiota populations in *P. xylostella* system (*Figure 6A, B*, *Figure 6—figure supplement 1*), but the mechanism needs to be studied in the future.

Disruption of the PM component drosocrystallin led to PM damage and larval death in *D. melanogaster* (*Kuraishi et al., 2011*). Here, we determined that DMNT repressed the expression of *PxMucin*, which encodes a component of *P. xylostella* PM structure. RNAi-based knockdown of this gene led to a defect in the PM structure, lower pupation rate, and larval death (*Figure 4*), suggesting that DMNT may damage the PM and affect pest growth by regulating several signaling pathways related to PM formation rather than through a simple physical interaction with the PM. However, we cannot rule out the possibility that DMNT may limit *P. xylostella* development in multiple ways, for instance, by affecting the expression of genes involved in larval development or immunity (*Figure 4—figure supplement 1*), which will be an important research direction in the future to elucidate the biological function of DMNT.

In conclusion, we described the interplay between the natural metabolite DMNT, midgut microbiota, and insect host death. The ability of DMNT to affect the PM structure and kill insect pests requires the assistance of the microbiota from the insect midgut and may represent a general phenomenon in other organisms that explains how the intestinal microbiota is prevented from infecting their hosts.

## Materials and methods

**Key resources table**

| Reagent type (species) or resource | Designation | Source or reference | Identifiers | Additional information |
|---|---|---|---|---|
| Gene (*Arabidopsis thaliana*) | PEN1 | *Sohrabi et al., 2015* | At4g15340 (Tair) | *35Spro:PEN1-Nos* construction |
| *Gene (Plutella xylostella)* | *PxMucin* | *Sarauer et al., 2003*; *Israni and Rajam, 2017* | LOC105381635(NCBI) | qRT-PCR assay |
| Strain, strain background (*Enterococcus* sp.) | *Enterococcus faecalis (E. f)* | This paper | ATCC29212 (http://www.bncc.org.cn/) | Gram staining |
| Commercial assay or kit | T7 RiboMAX express RNAi synthesis kit | This paper | Promega, catalog number P1700 | Double-stranded RNA synthesis |
| Commercial assay or kit | Lipase activity | This paper | Jiancheng Company (http://www.njjcbio.com, catalog: A054-1-1) | Lipase enzyme activity assay |

*Continued on next page*

*Continued*

| Reagent type (species) or resource | Designation | Source or reference | Identifiers | Additional information |
|---|---|---|---|---|
| Chemical compound, drug | Dye erioglaucine disodium salt | *Amcheslavsky et al., 2009*; *Rera et al., 2012*; *He et al., 2017* | Sigma-Aldrich, cas: 3844-45-9, molecular weight: 792.85, color index number 42090 | 'Smurf' treatment |
| Chemical compound, drug | Antibiotic cocktail | *Rodgers et al., 2017* | Solarbio, P1410 | Killing gut microbes |
| Chemical compound, drug | DMNT | *Huang and Yang, 2007* | | Chemical synthesis of DMNT |
| Software, algorithm | SPSS | SPSS | RRID:SCR_002865 | statistical analyses |
| Other | Polydimethylsiloxane solid phase micro-extraction (SPME) fibers | *Sohrabi et al., 2015* | 100 μm, DVB/CAR/PDMS, Supelco, Inc, Bellefonte, PA | Collection of volatile compounds |

## Plant materials and growth conditions

*A. thaliana* Col-0 seeds were ordered from Nottingham Arabidopsis Stock Centre (http://arabidopsis.info), and *B. napus* seeds were ordered from Wuhan Weimi Company. These plants were grown under a long-day photoperiod (16 hr light/8 hr dark) at 22℃. LED light intensity was set as 153 µmol m$^{-2}$ s$^{-1}$. Humidity was set at 70%. Also, 3-week-old *A. thaliana* seedlings and 5-week-old *B. napus* were used for bioassays.

## Generation of transgenic *Arabidopsis* plants overexpressing *PEN1*

The *PEN1* (at4g15340) coding region was amplified by PCR from a cDNA of the Col-0 accession and ligated into the pLGNL-35S vector after digestion with appropriate restriction enzymes. The vector places *PEN1* transcription under the control of the Cauliflower mosaic virus (CaMV) 35S promoter and uses the nopaline synthase (Nos) terminator. The *35Spro:PEN1-Nos* construct was introduced into *Agrobacterium* (*Agrobacterium tumefaciens*) strain GV3101 and transformed into Col-0 plants by the floral dipping method. We selected transformants using hygromycin (20 mg/mL) in Murashige and Skoog (MS) medium without sucrose, and the stable T3 lines were used for bioassays.

## GC-MS of DMNT in *35Spro:PEN1* transgenic plants

Three-week-old *A. thaliana* seedlings grown on Murashige and Skoog growth medium (MS-glucose) were used for DMNT measurement. The plant growth condition is: long-day photoperiod (16 hr light/8 hr dark), at 22℃. One-gram seedlings were weighed and transferred into 50 mL glass bottles. Volatile compounds were collected with polydimethylsiloxane solid phase micro-extraction (SPME) fibers (100 μm, DVB/CAR/PDMS, Supelco, Inc, Bellefonte, PA) for 1 hr at room temperature with a headspace absorption method. Volatile compounds were released by heating at 250℃ for 5 min and detected in gas chromatography (Agilent 7890A)-mass spectrometry (Agilent 5975C) platform. The machine was set as: air flow: 1 mL/min, temperature gradient: 5℃/min, from 40℃ to 220℃ (2 min hold), 20℃/min, from 220℃ to 240℃ (2 min hold). The separated compounds were analyzed with NIST11 mass spectrometer. DMNT was dissolved in methanol and analyzed with the same procedure as a comparison standard. Col-0 was used as a control for the analysis on *35Spro:PEN1* transgenic plants. To test the DMNT content in *P. xylostella*-infested plants, we used *35Spro:PEN1* transgenic plants that were infested with second-instar *P. xylostella* larvae for 3 to 21 hr.

## RNA-seq analysis

Total RNA from the whole body of second-instar *P. xylostella* larva was extracted with TRIzol RNA preparation method (Invitrogen, CA) and quantified and qualified with NanoDrop ND-1000 spectrometer (NanoDrop, Wilmington, DE). RNA intactness was inspected with Bioanalyzer 2100 (Agilent, CA) and Agarose electrophoresis method. The RNA samples with required quality (concentration >50 ng/µL, RIN >7.0, OD260/280 > 1.8, total RNA >1 µg) were used for further experiments. Poly(A) RNA was purified twice with oligo(dT) magnetic beads (Dynabeads Oligo(dT), catalog number 25-61005, Thermo Fisher). The purified mRNA was fragmented with NEBNextR

Magnesium RNA Fragmentation Module kit (catalog number E6150S) at 94°C for 5–7 min. The fragmented RNA was reverse-transcribed with Invitrogen SuperScript II Reverse Transcriptase (catalog number 1896649) to synthesize first-strand cDNA. *Escherichia coli* DNA polymerase I (NEB, catalog number m0209) and RNase H (NEB, catalog number m0297) were used to synthesize second-strand cDNA. During the synthesis, dUTP Solution (Thermo Fisher, catalog number R0133). cDNA was ligated to adaptors and then digested with UDG enzyme (NEB, catalog number m0280). DNA Library was prepared with PCR method. High-throughput sequencing was performed in Lianchuan Company with illumina Novaseq 6000 platform following normal DNA sequencing procedures (http://www.lc-bio.com/). The raw data generated by sequencing were preprocessed, using cutadapt to filter out unqualified sequences to obtain clean data, and then proceed to the next step of analysis. The specific processing steps are as follows: (1) remove the adapter (Adaptor) of reads; (2) remove reads that contain N (N means that the base information cannot be determined) more than 5%; (3) remove low-quality reads (the number of bases with a quality value $Q \leq 10$ accounts for 20% of the entire read above); (4) count the original sequencing amount, effective sequencing amount, Q20, Q30, GC content, and conduct a comprehensive evaluation. Then use Hisat to compare the preprocessed Valid Data with reference genome (https://www.ncbi.nlm.nih.gov/genome/11570?genome_assembly_id=40127). Reads Per Kilobase of exon model per Million mapped reads (RPKM) or Fragments Per Kilobase of exon model per Million mapped reads (FPKM) were used to measure the abundance of gene expression. Finally, we used edgeR to perform difference analysis on the genes assembled and quantified by StringTie (the threshold of significant difference is | log2foldchange| $\geq$ 1, p<0.05).

## Insects and bacteria

The eggs of *P. xylostella* were purchased from Henan Jiyuan Baiyun Company in China (http://www.keyunnpv.cn/) and incubated in a growth chamber set to 26°C, 16 hr light/8 hr dark. The hatched larvae were fed with commercial forage until a specific stage, as specified in the text, and transferred to bioassay tests.

*Enterococcus faecalis* ATCC29212 was purchased from the BeNa Culture Collection in China (http://www.bncc.org.cn/) and cultured in growth medium (in 100 mL medium: 0.3 g beef paste, 1.0 g peptone, and 0.5 g NaCl) at 37°C according to the company's instructions.

## Preference test of *P. xylostella* larvae upon DMNT treatment

We designed an in-house insect preference test device to measure the preference of *P. xylostella* larvae (*Figure 1—figure supplement 2*). Forage, containing the indicated amounts of DMNT and the solvent paraffin oil, was placed at each end of the glass test tube. Along with the air flow generated by a pump (1 mL/min), we placed 15 *P. xylostella* larvae in the center of the test tube for each assay. After 10–120 min, the number of larvae that had moved to either end of the glass tube were scored and compared to determine larvae preferences. To avoid any influence from the experimental setup, we swapped the DMNT and control paraffin oil positions in half of our assays. All assays were repeated at least six times with consistent results.

## No-choice test of *P. xylostella* larvae

Except when testing different concentrations of DMNT, all bioassays in this study used 7 µM DMNT and second-instar *P. xylostella* larvae. For each assay, we performed six independent replicates, each consisting of 15 larvae. The experiments were carried out in clear Petri dishes wrapped with surgical tape (3M Micropore) for air ventilation. Within each dish, we placed three forage blocks (diameter 1.5 cm, height 0.8 cm) and added 15 µL of 0.1 mg/mL DMNT dissolved in paraffin oil (Sigma) as solvent on top of each block to bring the final DMNT concentration in forage to 7 µM. The same volume of paraffin oil was used as controls. After different lengths of DMNT exposure, the survival, body weight, pupation, and eclosion data were collected and compared.

## Lipase enzyme activity assay in *P. xylostella* larva midguts

The entire midguts of 90 *P. xylostella* larvae were isolated by hand and quickly frozen in liquid nitrogen. Lipase activity was assayed with a commercial kit from Jiancheng Company (http://www.njjcbio.com, catalog: A054-1-1) according to the manufacturer's instructions with a spectrophotometer.

Briefly, the substrate buffer was warmed to 37°C before the assays. The midguts of *P. xylostella* larvae were homogenized in 25 µL homogenization solution and transferred into 25 µL buffer #4 and 2 mL pre-warmed substrate buffer. The mixture was quickly moved to a spectrophotometer to check the absorbance (A1) at 420 nm. The mixture was then transferred to a water bath set to 37°C for incubation. After 10 min, the mixture was transferred to the spectrophotometer to measure the absorbance (A2). As a control, 2 mL substrate solution was mixed with 50 µL 0.9% NaCl solution for absorbance measurement at 420 nm (As). Enzyme activity was calculated according to the equation: [(A1–A2)/As]×(454 µmol/L)×[reaction volume (2.05 mL)/sample volume (0.025 mL)]/reaction time (10 min)/protein concentration of sample (gprot/L).

### Starvation test of *P. xylostella* larvae

Second-instar *P. xylostella* larvae were selected randomly for DMNT and starvation tests in separated Petri dishes. DMNT dissolved in paraffin oil was injected into forage, and the same amount of paraffin oil was put into the Petri dishes for starvation test. After 12–36 hr, the mortality of *P. xylostella* larvae was monitored and recorded.

### Antibiotic cocktail treatment of *P. xylostella* larvae

We prepared a cocktail of antibiotics by mixing penicillin, streptomycin, and gentamycin at the working concentration of 1 kU/mL, 1 mg/mL, and 0.5 mg/mL, respectively. Forage blocks were immersed in an antibiotic cocktail solution for 30 s, air-dried and fed to second-instar *P. xylostella* larvae with or without DMNT in specific experiments. To test whether bacteria were killed by the antibiotic cocktail, we dissected the midgut from *P. xylostella* larvae 72 hr after treatment under sterile conditions and transferred the tissues into 1 mL LB medium for a 48 hr incubation at 37°C. The 30 µL culture was then spread onto LB plates containing no antibiotics for growth at 37°C for 48 hr, after which we determined bacterial growth. Before dissecting guts, the surface of whole larva body was disinfected with 75% ethanol. To prepare second-instar 'antibiotics-treated' *P. xylostella* larva caterpillars, we first inoculated larval eggs on plastic paper and put it close to forage containing antibiotics to ensure the larvae could reach and eat antibiotics immediately after hatching. To test whether bacteria were killed by the antibiotic cocktail, we dissected the midgut from second-instar *P. xylostella* larvae and transferred the tissues into 50 µL LB medium, then spread it on LB solid medium for incubation overnight at 37°C. All experiments were performed in a sterile hood to avoid contamination. As control, we performed the same experiment with *P. xylostella* larvae not exposed to antibiotic treatment.

### Smurf assay of *P. xylostella* midgut permeability

The assay was carried out using the dye erioglaucine disodium salt (Sigma-Aldrich, cas 3844-45-9, molecular weight 792.85, color index number 42090) as described with some modifications (*Amcheslavsky et al., 2009*; *Rera et al., 2012*; *He et al., 2017*). Briefly, third-instar *P. xylostella* larvae were fed with forage containing DMNT for 48 hr. Forage was then replaced with fresh forage containing 0.3125 mg/mL dye, and larvae were allowed to feed for 1 hr before being given fresh forage with no added DMNT or dye. After 6 hr, the larvae were photographed under a stereomicroscope. Six biological replicates were prepared for the treatment and control, with 15 larvae per replicate. When the midguts are intact, the dye will pass through together with feces; if the midguts are damaged, the dye will penetrate into larval tissues and remain inside the body, thus turning larvae blue (hence the name of the assay).

### Effects of intestinal microbiota on the PM of *P. xylostella* larvae

To prepare the gut microbes, we dissected the midgut from 45 *P. xylostella* larvae and transferred the tissues into 200 mL medium incubation at 37°C. After 48 hr, we collected the gut microbes by centrifugation at 2841 *g* for 15 min and then added 1×PBS (phosphate buffer saline) for later use. Before dissecting guts, the surface of whole larvae was disinfected with 75% ethanol. All experiments were performed in a sterile hood to avoid contamination.

## Dissection of midgut and PM of *P. xylostella* larvae

*P. xylostella* larvae were narcotized by ice incubation for 30 min, and then immersed in sterile 1×PBS solution (pH 7.2). The entire midguts were dissected and one end of midgut was opened with an anatomical needle. The other end of the midgut was handled with a forceps and gently shaken in the PBS solution, allowing the PM to float out. PM features were inspected and imaged under a Leica stereoscope.

## Transverse sections of *P. xylostella* larvae

*P. xylostella* larvae were fixed in 1% formaldehyde in 1×PBS under gentle vacuum and later embedded in paraffin, following dehydration with a graded ethanol series. Individual larvae were sectioned on a microtome at the middle position of the body and de-waxed with a series of xylene solutions before staining with HE staining solution on slides (*Rodgers et al., 2017*). The slides were then scanned to obtain images on a high-resolution scanner (Pannoramic Scanner, DESK).

## Chemical synthesis of DMNT

DMNT was synthesized according to published literature with minor modifications (*Huang and Yang, 2007*). The purity of all batches of DMNT was determined by nuclear magnetic resonance.

## Sequencing of 16S ribosomal DNA from midgut microbiota

After feeding for 72 hr, *P. xylostella* larvae were dissected to collect the content of their midguts. Before dissecting guts, the surface of whole larva body was disinfected with 75% ethanol. Eight independent biological replicates were performed, each comprising 15 *P. xylostella* larvae. Gut microbe DNA was extracted with E.Z.N.Z Stool DNA Kit (D4015, Omega, lnc.). The conserved regions of 16S ribosomal DNA were amplified by PCR, separated in 2% agarose gels, purified with AMPure XT beads (Beckman Coulter Genomics, Danvers, MA), and quantified by Qubit (Invitrogen). The amplicon pools were prepared for sequencing and the size and quantity of the amplicon library were assessed on Agilent 2100 Bioanalyzer (Agilent) and with the Library Quantification Kit for Illumina (Kapa Biosciences, Woburn, MA), respectively. The libraries were sequenced on NovaSeq PE250 platform.

## Data analysis

Samples were sequenced on an Illumina NovaSeq platform according to the manufacturer's recommendations, provided by LC-Bio. Paired-end reads were assigned to samples based on their unique barcode and truncated by cutting off the barcode and primer sequence. Paired-end reads were merged using FLASH. Quality filtering on the raw reads was performed under specific filtering conditions to obtain the high-quality clean tags according to the fqtrim (v0.94). Chimeric sequences were filtered using Vsearch software (v2.3.4). After dereplication using DADA2, we obtained the feature table and feature sequence. Alpha diversity and beta diversity were calculated by normalizing to the same sequences randomly. Then according to SILVA (release 132) classifier, feature abundance was normalized using relative abundance of each sample. Alpha diversity is applied in analyzing the complexity of species diversity for a sample through five indices, including Chao1, Observed species, Goods coverage, Shannon, and Simpson, and all these indices in our samples were calculated with QIIME2. Beta diversity was calculated by QIIME2, and the graphs were drawn by R package. Blast was used for sequence alignment, and the feature sequences were annotated with SILVA database for each representative sequence. Other diagrams were implemented using the R package (v3.5.2). This work was performed by Lianchuan Company (http://www.lc-bio.com/).

## Statistical analysis

All statistical analyses in this study were performed in the SPSS 16.0 package as two-tailed unpaired *t*-tests, ANOVA, or Kaplan–Meier survival analysis using default parameters with no data transformation (http://www.spss.com). Replication numbers for all assays are listed in the figure legends.

## dsRNA synthesis and treatment

dsRNA was synthesized with the T7 RiboMAX express RNAi synthesis kit (Promega, catalog number P1700) as described in the manufacturer's instructions. The primers for RNAi of *PxMucin* (LOC105381635) are listed in *Supplementary file 2*, and the length of dsRNA is 502 bp.

To test the function of *PxMucin* on the growth of *P. xylostella* larvae, 20 μL *PxMucin* dsRNA (500 ng/μL) was spread onto the leaves of 2-week-old *A. thaliana* plants, which were used to feed first-instar *P. xylostella* larvae. A dsRNA for the green fluorescent protein (GFP) was used as a negative control, and the length of dsRNA is 463 bp. GFP fragment was amplified with PCR method using pCambia-1305GFP plasmid as DNA template. After feeding for 24–96 hr, the survival, weight, and pupation rates were scored for all larvae. Six biological replicates were set up for each treatment, each replicate consisting of 15 larvae. After the larvae had fed for 24 hr, the midgut tissues were dissected for RNA extraction using Trizol method, reverse transcription with one-step cDNA synthesis kit (Takara), and *PxMucin* expression analysis by quantitative PCR techniques (Roche, Light Cycler 480C), three biological replicates were performed, with each biological replicate including 30 larvae. The positions of the qRT-PCR analysis primers are as shown in *Figure 4A*, with the forward primer on exon 7 and the reverse primer in intron 7. The primer sequences are listed in *Supplementary file 2*. In addition, after 48 hr of feeding, larvae were processed for sections and HE staining to observe the effects of *PxMucin* RNAi on PM.

## qRT-PCR analysis

Total RNA of *P. xylostella* larva midgut was extracted with TRIzol (TaKaRa) RNA preparation method, and 1 μg RNA was used for reverse transcription to synthesize cDNA. The specific primers are listed in *Supplementary file 2*. *PxRPL32* and *PxRPS13* were used as reference genes (*Fu et al., 2013*). The primers for *PxLysozyme* were designed by *Xia et al., 2018*.

# Acknowledgements

We thank the members of Professor Tang's group for assistance and helpful instructions on the bioassays. We also thank the members from Li lab for valuable discussion. This work was supported by the National Key Research and Development Program of China (2017YFD0301301, 2016YFD0101803) and the Natural Science Foundation of China (31670264). The authors declare no conflict of interest.

# Additional information

### Funding

| Funder | Grant reference number | Author |
| --- | --- | --- |
| National Key Research and Development Program of China | 2017YFD0301301 | Peijin Li |
| National Key Research and Development Program of China | 2016YFD0101803 | Peijin Li |
| National Natural Science Foundation of China | 31670264 | Peijin Li |

The funders had no role in study design, data collection and interpretation, or the decision to submit the work for publication.

### Author contributions

Chen Chen, Conceptualization, Data curation, Formal analysis, Validation, Investigation, Visualization, Methodology; Hongyi Chen, Shijie Huang, Formal analysis, Investigation; Taoshan Jiang, Chuanhong Wang, Zhen Tao, Formal analysis, Methodology; Chen He, Qingfeng Tang, Formal analysis; Peijin Li, Conceptualization, Resources, Formal analysis, Writing - original draft, Writing - review and editing

## Author ORCIDs

Peijin Li  https://orcid.org/0000-0003-1579-7553

## Decision letter and Author response

Decision letter https://doi.org/10.7554/eLife.63938.sa1
Author response https://doi.org/10.7554/eLife.63938.sa2

## Additional files

### Supplementary files

- Supplementary file 1. Differentially expressed genes from RNA-seq.
- Supplementary file 2. Primers used in this study.
- Supplementary file 3. Abundance of major Operational Taxonomic Units (OTUs).
- Transparent reporting form

### Data availability

All data generated or analysed during this study are included in the manuscript and supporting files. Source data files have been provided for Figure 1–6, Figure 1-figure supplement 1, 3–5, Figure 4-figure supplement 2, Figure 5-figure supplement 2, and Figure 6-figure supplement 2,3.

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
