## [Decision Letter]

**Acceptance summary:**

This paper provides the new role of plant volatiles which directly damages the insect gut system through the gut microbiome of insects. The study nicely genetically manipulates a plant and an insect to reveal the mode of action of the direct defensive role of plant volatiles. Therefore, this work provides important insights into the new layer of plant defense and future agricultural applications.

**Decision letter after peer review:**

Thank you for submitting your article "Volatile DMNT protects plants against *Plutella xylostella* by disrupting the peritrophic matrix barrier in insect midgut" for consideration by *eLife*. Your article has been reviewed by three peer reviewers, including Youngsung Joo as the Reviewing Editor and Reviewer #1, and the evaluation has been overseen by Meredith Schuman as the Senior Editor.

The reviewers have discussed the reviews with one another and the Reviewing Editor has drafted this decision to help you prepare a revised submission.

Summary:

This paper is of potential interest to broad audience of plant science and pest management. It provides evidence for a new role of plant volatile compounds which directly damages the insect gut system, dependent on the microbiome. All three reviewers agreed that this manuscript has the potential for publication in *eLife*, but four major issues need to be solved before acceptance.

Essential revisions:

1) There are concerns about the ecological relevance of DMNT treatment:

a) Why did the authors use 7 μm DMNT throughout the experiment? The authors found little difference in effects across 3 orders of magnitude (0.7, 7, 70 uM, Figure 1—figure supplement 3). How does this dose compare to the estimated dose which larvae receive when feeding on plant tissue?

b) The authors already confirmed that the role of DMNT depends on native gut microbiota in Figure 5, so why have *Enterococcus faecalis* and *E. coli* has been selected to test the DMNT role in the gut microbiome? Why didn’t the authors use cultured bacteria from the insect gut? These issues should be addressed in the manuscript.

2) There is a significant concern on whether the gut bacteria in the caterpillars are required for the negative effects of DMNT as plant resistance. Particularly, reviewer #3 is concerned that bacteria-free (axenic) caterpillar cannot be replaced by antibiotic treatment. I have discussed with the reviewer and I think that antibiotic treatment is still sufficient to support the gut microbiota (antibiotics-sensitive)-dependent effects of DMNT. However, the external pathogen inoculation experiment in Figure 6 should be done using "antibiotics-treated" caterpillars to demonstrate that gut bacteria are required for DMNT-mediated negative effects on the larvae.

3) Besides, the effects of DMNT between Figure 5 and 6 significantly differ. It seems that DMNT effects are significantly dampened in Figure 6. This, even there is no difference in the survival of larvae between control and DMNT treatment at 60h and 84h. Is there any reason causing this difference? Maybe it is also caused by the batch effects among experiments. All X-axis of larval survival graph in Figure 5A, C and Figure 6C, E are different. It is necessary to address why the effects of DMNT largely differed throughout the manuscript (despite a consistent dose).

4) Lastly, the authors need to introduce and discuss their findings with other relevant literature.

Introduction

The defensive role of plant volatile has not been studied, so this is one of the main novelties of this study. So, it is necessary emphasize this point in the Introduction by citing previous studies like Maurya et al., 2020, Gasmi et al., 2019, etc. Also, it would be better to change the title to "Volatile DMNT directly protects plants against *Plutella xylostella* by disrupting the peritrophic matrix barrier in insect midgut".

Discussion

The findings need to be discussed with other relevant literature. Several studies showed that plant volatiles are directly toxic to caterpillars or change caterpillar physiology. Here are some examples:

– Sugimoto et al., 2014;

– Veyrat et al., 2016;

– Ye et al., 2018.

---

## [Author Response]

Essential revisions:1) There are concerns about the ecological relevance of DMNT treatment:a) Why did the authors use 7 μm DMNT throughout the experiment? The authors found little difference in effects across 3 orders of magnitude (0.7, 7, 70 uM, Figure 1—figure supplement 3). How does this dose compare to the estimated dose which larvae receive when feeding on plant tissue?

Thank you very much for these valuable comments, we agree and have added additional data for DMNT quantification in *Arabidopsis thaliana* plants.

Based on the report from Soharbi et al., 2015.DMNT content in *Arabidopsis thaliana* tissues can be induced to ~70 ng/g after 100 mM JA treatment. We set up 3 dosages to test the influence of DMNT on the development of second instar *P. xylostella* larvae, 0.7 μM (65 ng/g), 7 μM (650 ng/g) and 70 μM (6.5 μg/g) respectively, which showed similar effect on larvar survival and pupation (Figure 1—figure supplement 3). Although the dosage 65 ng/g is close to the DMNT content in *Arabidopsis* tissues, we selected a higher dosage for bioassays, as we thought that alive plants could continuously produce DMNT and the abundance in plants will be upregulated and accumulated upon larva attacks; but in contrast in bioassays, the DMNT injected in forage will become less and less because of volatilizing and being eaten by *P. xylostella* along with the process of experiments. Indeed, consistent with this prediction, when *35Spro:PEN1* transgenic *Arabidopsis thaliana* plants were infested by *P. xylostella* for 3 to 21 hours, DMNT content could rise from 10 to 68 ng/g. We have added the additional data in the Figure 1—figure supplement 1C, D. Please refer to the Results and Discussion sections.

b) The authors already confirmed that the role of DMNT depends on native gut microbiota in Figure 5, so why have Enterococcus faecalis and E. coli has been selected to test the DMNT role in the gut microbiome? Why didn’t the authors use cultured bacteria from the insect gut? These issues should be addressed in the manuscript.

We are grateful for these comments to use the cultured bacteria from insect gut instead of *Enterococcus faecalis* or *E. coli*, which will mimic the reality of gut environment much better. Using the method described by Wang et al. (Plant Pathogens, 2020, C-type lectin-mediated microbial homeostasis is critical for *Helicoverpa armigera* larval growth and development), we isolated and cultured the native microbes from gut for the treatment on *P. xylostella*, the results showed that the native microbes presented much stronger effect than *E. coli* or *Enterococcus faecalis* alone, all larvae died at 120 h and no larvae pupated in the end. These new results have been added to the Results section.

2) There is a significant concern on whether the gut bacteria in the caterpillars are required for the negative effects of DMNT as plant resistance. Particularly, reviewer #3 is concerned that bacteria-free (axenic) caterpillar cannot be replaced by antibiotic treatment. I have discussed with the reviewer and I think that antibiotic treatment is still sufficient to support the gut microbiota (antibiotics-sensitive)-dependent effects of DMNT. However, the external pathogen inoculation experiment in Figure 6 should be done using "antibiotics-treated" caterpillars to demonstrate that gut bacteria are required for DMNT-mediated negative effects on the larvae.

We agree with these comments and have performed new bioassays as suggested. Thank you very much.

To prepare second instar “antibiotics-treated” *P. xylostella* larvae, we firstly inoculated larval eggs on plastic paper and put it close to forage containing antibiotics to ensure the larvae could eat antibiotics immediately after hatching. As shown in the new figure 6—figure supplement 3, the gut microbiota from antibiotics-treated *P. xylostella* larvae at second instar stage was nearly killed by antibiotics.

We got consistent results as those using the *P. xylostella* larvae that had not been pre-treated with antibiotics shown in the last version. When using “antibiotics-treated” *P. xylostella* larvae for bioassays, addition of gut microbiota in forage showed slightly but significantly restraint on larva growth as expected, and DMNT+antibiotics treatment showed very little effect; whereas in contrast, the treatment of DMNT+gut microbes showed very strong influence, all the larvae died 120 hours after treatment, and no larvae pupated, suggesting gut microbiota is essential for the killing function of DMNT (Figure 6C, D).

The new bioassay method for the preparation of “antibiotics-treated larvae” has been added to the Materials and methods section and these new results have been added in the Results section. Please refer to the new Figure 6C, D, Figure 6—figure supplement 3.

3) Besides, the effects of DMNT between Figures 5 and 6 significantly differ. It seems that DMNT effects are significantly dampened in Figure 6. This, even there is no difference in the survival of larvae between control and DMNT treatment at 60h and 84h. Is there any reason causing this difference? Maybe it is also caused by the batch effects among experiments. All X-axis of larval survival graph in Figure 5A, C and Figure 6C, E are different. It is necessary to address why the effects of DMNT largely differed throughout the manuscript (despite a consistent dose).

We are sorry we did not explain the bioassay method clearly. For Figure 5, DMNT was added to forage from the start to the end of the bioassays, whereas for Figure 6, to reveal the effect of bacterial on the larvae with damaged gut by DMNT, we used the forage containing DMNT to feed *P. xylostella* for only 48 h, then the forage was replaced with the forage with *E. coli* or *E. faecalis*, but without DMNT, so the overall treatment effect in Figure 6 is relatively less pronounced than that in Figure 5.

According to the reviewers’ comments, we have performed new experiments using the native bacterial from larval gut in Figure 6C, D. Please refer to the new Figure 6C, D and the Results section. Additionally, we corrected and used consistent time points, to ensure the X-axis between Figures 5 and 6 are the same. Please refer to the new Figures 5 and 6.

4) Lastly, the authors need to introduce and discuss their findings with other relevant literature.

Thank you very much. We agree and have added more introduction and discussion as suggested.

IntroductionThe defensive role of plant volatile has not been studied, so this is one of the main novelties of this study. So, it is necessary emphasize this point in the Introduction by citing previous studies like Maurya et al., 2020, Gasmi et al., 2019, etc. Also, it would be better to change the title to "Volatile DMNT directly protects plants against Plutella xylostella by disrupting the peritrophic matrix barrier in insect midgut".

Thank you. We agree and have added more content in the Introduction section to describe the function of plant volatile as direct toxins by describing and citing the researches from Maurya, Gasmi et al.

We appreciate and agree to use the new title as the reviewers recommended, which could summarize our findings better.

DiscussionThe findings need to be discussed with other relevant literature. Several studies showed that plant volatiles are directly toxic to caterpillars or change caterpillar physiology. Here are some examples:– Sugimoto et al.. 2014;– Veyrat et al., 2016;– Ye et al., 2018.

We are grateful for these valuable comments. We agree and have updated the Discussion parts by including the relevant literature to compare their findings with our results.